# Parameter Optimization for an Accurate Swept-Sine Identification Procedure of Nonlinear Systems

**Pietro Burrascano** 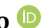

Dipartimento di Ingegneria, Università di Perugia, 06125 Perugia, Italy; pietro.burrascano@unipg.it

**Abstract:** Modeling the nonlinearity of a system is of primary importance both for optimizing its design and for controlling the behavior of physical systems operating with a wide dynamic range of input values, for which the linearity hypothesis may not be sufficient. To become of practical use, the identification of nonlinear models must be accurate and computationally efficient. For these reasons, in recent years, among the numerous models of nonlinear systems that have been proposed in the technical literature, the Hammerstein model has been widely applied as a consequence of the proposal of a new pattern identification technique based on pulse compression, which makes the identification of the model very accurate in numerous applications for which it has been adopted. Hammerstein model identification of a nonlinear system requires characterization of the linear filters present on the different branches of the model. These linear filters, which constitute the parameters of the model to be identified, must be considered with respect to their trends over time or, equivalently, in their frequency trends, as amplitude and phase responses. The identification can be considered accurate if the trends obtained for each filter adequately characterize it for the entire frequency range to which that specific filter is subjected in the normal operation of the system to be identified. This work focuses on this aspect, i.e., on the adequacy of the frequency range for which the filter is identified and on how to obtain correct identification in the entire frequency range of interest. The identification procedure based on exponential swept-sine signals defines these filters in the time domain by making use of intermediate functions that are related to the impulse responses of the model filters through a linear transformation. In this paper, we analyze, in detail, the roles of the bandwidths of both the excitation signal and the matched filter, which are the basis of the procedure, we verify the assumptions made about the amplitudes of their frequency bands, and we propose criteria for defining the bandwidths in order to maximize accuracy in model identification. The experiment performed makes it possible to verify that the proposed procedure avoids possible limitations and significantly improves the quality of the identification results, both if the description is made in the time domain and in the frequency domain.

**Keywords:** nonlinear systems; pulse compression; Hammerstein model identification; bandwidth limitation effects

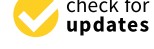



## 1. Introduction

A number of different modeling techniques that are capable of representing the nonlinear behavior of physical devices have been proposed and, for each of these, there are sophisticated techniques for identifying the parameters that characterize the modeling technique.

Modeling techniques of nonlinear devices can be grouped into three main categories that differ in the level of knowledge of the physical phenomenon that is represented in the model itself. The so-called white-box approaches require complete knowledge of the physics governing the nonlinear system [1]. Examples of these techniques are those that define the model of the real physical system by making use of differential equations [2] or wave digital filters [3,4].

The complexity of a physical system often makes it necessary to carry out simplification techniques. Depending on the degree of simplification, the techniques are considered to

be part of the grey box or black box approaches. The grey box techniques imply partial knowledge of the physical phenomenon [5,6], for example, in [7,8]; the black box techniques, which in fact are among the most widely adopted, do not require prior knowledge of the physics of the system and the device is defined through its input–output relation [1,9]. In the following, only nonlinear systems of the black box type are considered.

In the case of black box techniques, the most widely adopted model is the Volterra series [10,11]. For practical reasons, the series must be truncated to bring the number of model parameters to a finite number. Even in the case of a truncated Volterra series, the number of coefficients needed to define the model quickly becomes very large as the degree of the model increases. This becomes a serious limitation in that, in practice, it allows this technique to be used only with systems characterized by a limited degree of nonlinearity. Simplified models with respect to the Volterra series have been studied to deal with cases of strong nonlinearities. Among them, the Hammerstein and Wiener models are based on a split between a part of the model representing the dynamics of the response through linear filters, and a part of the model representing the nonlinear part of the response, which is considered static. In this way, these models provide accurate, though less general, representations of nonlinear systems even in the case of high-degree nonlinearities [12,13].

There are numerous techniques in the technical literature for identifying such models [14,15], and in particular, in the case of the Hammerstein model, a technique that has proven to be particularly effective is based on the use of appropriate swept-sine signals as input [16–19]. This technique has yielded excellent results in numerous application areas, including acoustics and nondestructive testing and evaluation [20–23].

The excellent results achieved with this identification technique are, however, always related to integral analyses performed in the frequency domain. The results of this identification method appear to be less brillant with regard to the time behavior of the model response, or equivalently of the impulse response of the filters present in the different branches associated with the different orders of the model. Anomalies in the time response are often found, among which the most frequent is related to oscillations at transitions.

Artifacts of this type are known in the technical literature as Gibbs artifacts, and they manifest themselves in the form of spurious oscillations in the time domain response. It is well known that the cause of the occurrence of such spurious oscillations comes from bandwidth limitations of the performed measurements [24,25].

In the present paper, the focus is on the optimization of the Hammerstein's model identification procedure for the purpose of the optimization of its results as seen in the time domain. To this end, the entire identification procedure is revisited in order to verify the adequacy of the frequency band of each of the components that contribute to the identification procedure. A possible cause of criticality is identified in the choice of the parameters characterizing both the swept-sine signal used as excitation and the corresponding matched filter; the consequent criteria for choosing these parameters to optimize the response of the model even in the time domain are hypothesized.

The proposed solution is verified through experimental tests in simulation; the tests are defined by making use of a simulated nonlinear system so that the expected ideal response is known, and this allows the effectiveness of the proposed solution to be verified. The results of these experiments fully confirm the hypothesis made; consequently, in this paper, we provide clear guidance for choosing the signal parameters to be used in applying the identification procedure.

This paper is organized as follows: In Section 2.1, the Hammerstein's model of nonlinear systems is briefly described and its identification procedure based on the use of exponential swept-sine signals as input is presented. In Section 2.2, we analyze the identification procedure based on swept-sine signals in the frequency domain and we highlight the features of the identification procedure that have the potential to lead to frequency limitations in the functions describing the responses of the individual filters of the Hammerstein model. We then identify the possible causes of the above limitations and propose a possible solution. In Section 3, we define an experiment to test the reliability of the assumption

made about the causes of the limitations in the identification procedure and to identify the characteristics that the exponential swept-sine input signal is required to possess in order to enable accurate characterizations, in both the time and frequency domains of the linear filters present in the Hammerstein model to be identified. In Section 4, we discuss the results obtained. In Section 5, we draw conclusions and indicate possible evolution of the work.

## 2. Materials and Methods

A system for which the superposition principle is not valid in the relationship between input and output cannot be modeled through the usual methods of representing linear systems, and needs specific models. Historically, the model proposed by Volterra was the first among the representation models of nonlinear systems proposed in the technical literature [10,11]. It can be interpreted as an extension of convolution, i.e., of the input–output relation representing linear systems. The Volterra model, even for minor nonlinearities, is characterized by a very large number of parameters, which makes it extremely complex to define their values that fit the representation of a specific nonlinear system to be identified.

Block-structured models, such as the Wiener model, the Hammerstein model, and their combination, have been proposed as they imply a much lower number of parameters. The Wiener model consists of the sequence of a linear system with memory placed before the nonlinearity of the static type. The Hammerstein model is the reverse of the Wiener model. It consists of a static nonlinearity followed by a dynamic filter representing the memory of the system. The so-called $N_H$-order cascade of Hammerstein models also belong to this family, which consists of paralleling $N_H$ structures, each consisting of a static nonlinearity followed by a dynamic filter. This is the structure shown in Figure 1, and it is the one to which the paper refers in the following section. The identification methods are procedures for defining the parameters of the chosen model in such a way that the model's input–output behavior is equivalent to that of the nonlinear physical system it is intended to represent.

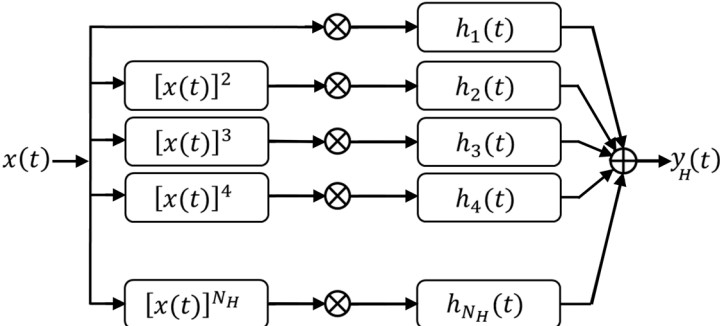

**Figure 1.** The Hammerstein model.

### 2.1. Hammerstein Model and Pulse-Compression Identification Procedure

Assume that the representation of the nonlinear system is validly carried out through a Hammerstein model of order $N_H$, the schematic description of which is shown in Figure 1. The physical system is designed to operate in the frequency band between $f_{MIN}$ and $f_{MAX}$.

A specific technique must be followed to identify the Hammerstein structure filter kernels that allow the model to have behavior equivalent to that of the nonlinear system being modeled. Among the identification techniques, those based on correlation involve using a specific input signal to the nonlinear system and analyzing the corresponding outputs of the system to define the individual filter kernels that identify the model. The identification technique based on pulse compression (PuC) belongs to this category [16–19]; it relies on the ability to define a couple of signals, $x(t)$ and $\psi(t)$, of which one represents the excitation signal and the other the impulse response of the matched filter, respectively. They

are defined such that the convolution between the two is the best possible approximation of the mathematical pulse:

$$o(t) = x(t) \otimes \psi(t) \cong \delta(t) \tag{1}$$

Since the signals have to be necessarily band limited, their convolution can only approximate the mathematical pulse. If the two signals are chosen in such a way that, in the frequency band from $f_{MIN}$ to the frequency $f_{MAX}$, the transform of their convolution has a nearly constant magnitude, the time course of this convolution will be of the type [23]:

$$f(t) \propto f_{MAX} \sin[2\pi f_{MAX}t]/[2\pi f_{MAX}t] - f_{MIN} \sin[2\pi f_{MIN} t]/[2\pi f_{MIN} t] \tag{2}$$

For the input–output relation of the system, if defined in the time domain, the following expression will hold:

$$y_H(t) = x(t) \otimes h_1(t) + x(t)^2 \otimes h_2(t) + \ldots + x(t)^{N_H} \otimes h_{N_H}(t) \tag{3}$$

in which the symbol $\otimes$ denotes convolution. The $N_H$ impulsive functions $h_i(t)$ completely characterize the model, and therefore, model identification coincides with the identification of these functions. To apply the PuC procedure for identification, the swept-sine signal of unit amplitude can be adopted as input, described by $x(t) = Cos(\phi(t))$. In this case, the previous equation can be expressed in compact form as:

$$y_H(t) = \left[ [Cos(\phi(t))]^k \right]^T \otimes [h(t)] \tag{4}$$

where the entities in square brackets represent vectors; $[h(t)]$ is the vector of the different kernels $h_k(t)$, $k = 1, \ldots, N_H$; $\left[ [Cos(\phi(t))]^k \right]^T$ is the transpose of the vector of powers of the input signal.

The Chebyshev polynomials of the first kind allow the vector of powers of the cosine functions to be expressed through the harmonics of the same functions [26], therefore, the above expression can be rewritten in an alternative form as:

$$y_H(t) = \left[ [A_c]^{-1} Cos(k \, \phi(t)) \right]^T \otimes [h(t)] = [Cos(k \, \phi(t))]^T \otimes [g(t)] \tag{5}$$

in which $[A_c]$ is the matrix of coefficients of the Chebyshev polynomials of the first kind and the vector and $[g(t)]$ contains $N_H$ impulsive patterns directly related to the kernels $[h(t)]$, through the relation $[g(t)] = \left[ [A_c]^{-1} \right]^T [h(t)]$ [19,21]. A comparison of Relations (4) and (5) shows that the output of the Hammerstein model can be expressed alternatively through the sum of the convolutions between the powers of the input signal and the $[h(t)]$ kernels or through the sum of the convolutions between the harmonics of the cosine functions and the $[g(t)]$ kernels. The $[g(t)]$ kernels and the $[h(t)]$ kernels are related through a linear transformation.

The identification procedure is based on this correspondence between the vectors of the $[h(t)]$ and $[g(t)]$. Let the evolution law of the instantaneous frequency of the swept signal be the exponential type, that is, let $\omega(t) = d\phi(t)/dt = 2\pi f_{MIN} Exp(t/L)$, where the constant $L = T_0/ln(f_{MAX}/f_{MIN})$ describes how quickly the frequency changes over time between $f_{MIN}$ and $f_{MAX}$ in a signal whose duration is $T_0$; then, the kth harmonic of the input signal corresponds to a simple shift of the signal $x(t)$ by the quantity $\Delta t_k = L \, ln(k)$. In fact:

$$\begin{aligned} f(t + \Delta t_k) &= f_{MIN} \, Exp[\tfrac{t+\Delta t_k}{L}] = f_{MIN} \, Exp[\tfrac{t+L \, ln(k)}{L}] = \\ &= k \, f_{MIN} \, Exp[t/L] = k \, f(t) \end{aligned} \tag{6}$$

This property is depicted in Figure 2 and is the basis of the Hammerstein model identification procedure described in [16–19]. In the particular case where the signal used for pulse compression is a swept sine of the exponential type and if, in addition, the

instantaneous phase of the input signal respects some specific constraints, as described in [27], the compression filter, which is characterized by a frequency trend as a function of time that is exactly the same as that of the excitation swept-sine signal, fits not only with the input signal at instant $\Delta t_0$, but also at instants $\Delta t_k$ with its harmonics of order $k$ produced by the nonlinear system.

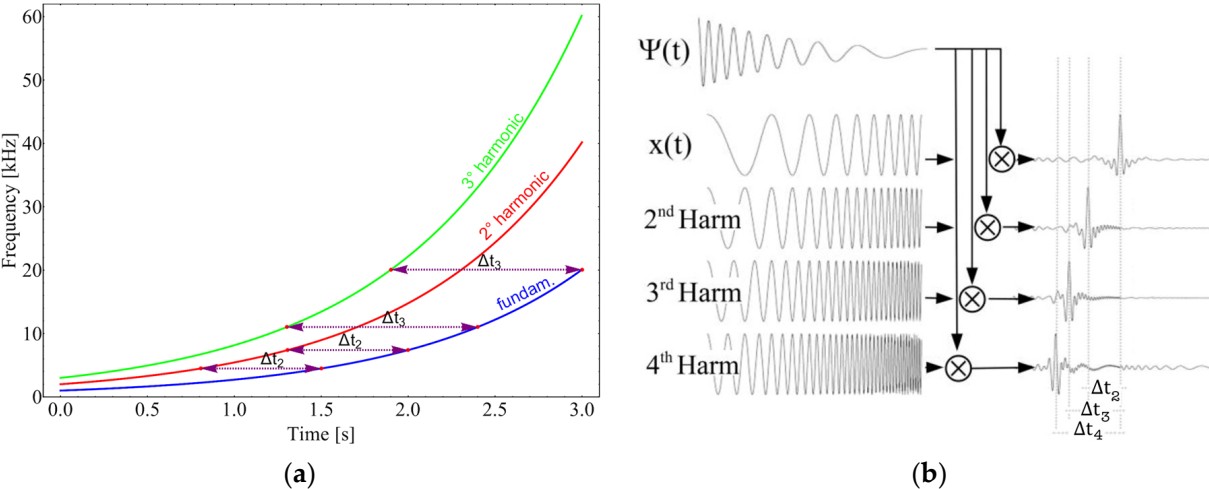

(**a**)                                              (**b**)

**Figure 2.** Exponential swept-sine signal at the output of a nonlinear device: (**a**) Time behavior of the fundamental frequency and of the first two harmonics. Note that the time distance between the frequency of a harmonic and the same frequency seen as the fundamental frequency is independent of the considered frequency; (**b**) time placement of the $g_k(t)$ functions at the output of the matched filter for the fundamental frequency and for the first harmonics.

The signal $y_H(t)$, processed through the matched filter $\psi(t)$, consists of a sequence of impulsive functions centered in the instants $\Delta t_k$; these impulsive functions coincide with the functions $g_k(t)$, as it is possible to write:

$$\mathbf{u}(t) = y_H(t) \otimes \psi(t) = [cos[k\phi(t)]_c]^T \otimes [g(t)] \otimes \psi(t) =$$
$$= \left\{ \left[ \hat{\delta}(t + \Delta t_k) \right]^T \right\} \otimes [g(t)] \tag{7}$$

where the $k$th element of the vector $\hat{\delta}(t)$ is the band-limited approximation of the Dirac delta function. Then, impulsive $g_k(t)$ waveforms are generated at such $\Delta t_k$ instants. Each of the $g_k(t)$ functions is associated with a specific harmonic generated by the nonlinear system, and the time positions of each $g_k(t)$ in the response of the matched filter indicate the order of the corresponding harmonic. In other words, the function $g_k(t)$ associated with the kth harmonic is present in the output signal to the matched filter and is shifted in time by an amount $\Delta t_k$ relative to the function $g_0(t)$ associated with the fundamental harmonic. Then, the functions $g_k(t)$ follow each other in the signal output to the matched filter; and then, each of the $g_k(t)$ functions is simply obtained by windowing the response of the matched filter at a time interval associated with the $k$th harmonic.

The PuC procedure, in all its phases up to the windowing required to obtain the $g_k(t)$ functions, is schematized in Figure 3. Once the $g_k(t)$ functions have been obtained by taking them at appropriate and known time instants of the signal u(t), the desired $h_k(t)$ functions can be obtained through the simple linear transformation $[h(t)] = [A_c]^T[g(t)]$.

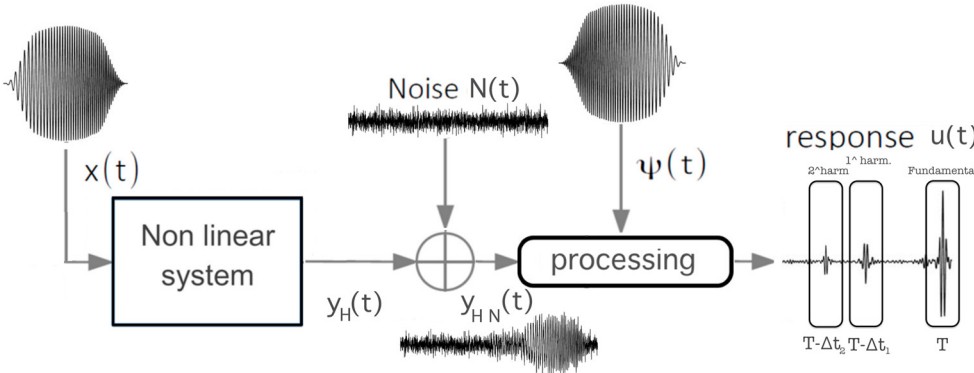

**Figure 3.** Schematic description of the PuC processing procedure for Hammerstein model identification.

## 2.2. Discussion on the Pulse-Compression Identification Procedure

In the first part of the present section, we verified, by analyzing the results of an experiment, the actual existence of the problem highlighted in the introduction, that is, the presence of spurious oscillations at the transitions, which alter the quality of the result, especially when considered in its trend over time. In the second part, the procedure leading to the identification of the patterns is analyzed and an interpretation of the highlighted problem is given, thus, also configuring a possible solution.

### 2.2.1. Limitations in the Identification Procedure

The pulse-compression identification technique has allowed excellent results to be obtained in numerous applications. For example, the results obtained in the estimation of intermodulation distortion carried out by means of a double exponential swept-sine signal [20], or in increasing the sensitivity of defect detection systems in materials by means of non-destructive techniques [21]. A careful analysis of the results, however, shows limitations that, while they do not affect the characteristics of the result in the case of integral-type estimator as, for example, in [21,22], they become significant in the case where there is interest in the time course of the response of the system represented through the nonlinear model. A verification of this statement can be achieved by analyzing the results of a laboratory experiment, the bench of which is shown in Figure 4. The experiment was performed to verify the presence of nonlinear effects in a couple of identical Tx and Rx air-coupled ultrasonic transducers.

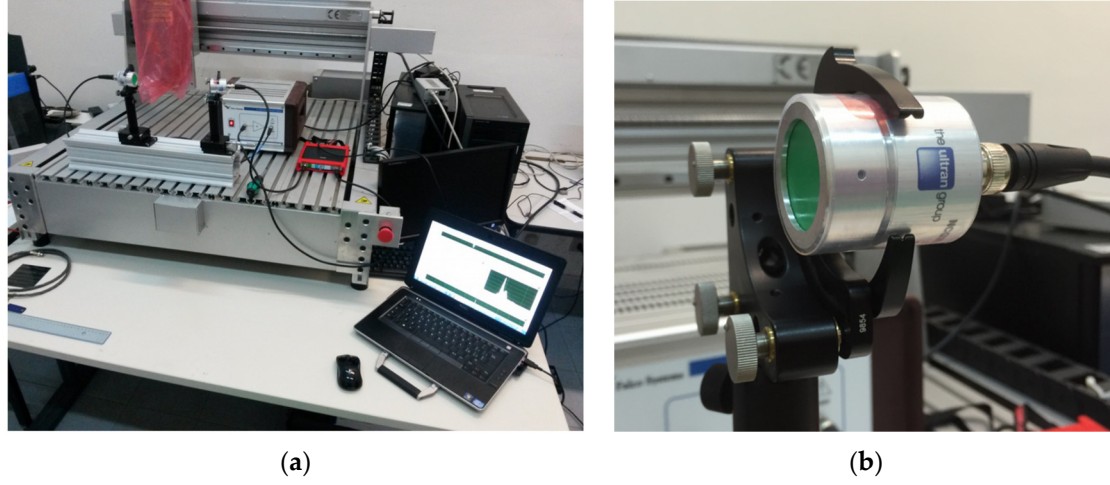

(**a**)                                                                                          (**b**)

**Figure 4.** The measurement bench (**a**) and one of the transducers (**b**) of the air-coupled ultrasonic transducers experiment.

The system configuration, shown in Figure 4, consists, in sequence, of a personal computer for measurement management and supervision, a TiePie handyscope HS5 [28] used as an arbitrary waveform generator, a Falco System power amplifier [29], a pair of identical Ultran brand transmit and receive transducers designed to emit in air [30], a TiePie handyscope HS5 used as a data logger, and a personal computer used for data storage.

The ultrasonic probes operate at a center frequency of 200 kHz and allow a maximum input voltage of 150 volts; both probes are of the focused type, with nominal focal distance F = 10 cm. The beam emitted in air by the transmitting piezoelectric transducer is focused at a single point (the focal point); the receiving system detects the signal emitted from a point on the axis of the transducer and placed at a focal distance of 10 cm. The Falco System signal amplification system is used on the signal generation side and has an amplification factor of $50\times$ in the useful band. The resolution adopted in the measurement was 14 bits. The ultrasonic probes were mounted on precision mounts and positioned at a distance of 21 cm, slightly more than twice the focal distance, which proved to be the distance at which the received signal level was found to be maximum.

Figure 5 shows the signal obtained at different stages of processing, starting from the data measured in the laboratory with the measuring bench described in Figure 4. Figure 5a shows the signal picked up by the receiving probe; Figure 5b plots the output of the matched filter, the figure also reports the positions in which the pulses corresponding to the different harmonics deriving from nonlinearity -if present- can be expected (dotted lines at $\Delta\delta_k = L\ ln(k)$); Figure 5c,d report a magnification of the curve in Figure 5b in the neighborhood of the time where the peaks corresponding to the fundamental frequency and to the second harmonic are expected.

Figure 5b,c highlight the aspect described in the Introduction, i.e., the presence of artifacts in the response over time, which manifest themselves in the form of spurious oscillations that are particularly evident in the instants preceding the ideal response attack. Possible causes of such limitations in the identification process will be analyzed in the following section.

2.2.2. PuC Identification Procedure Reformulated in the Frequency Domain to Highlight Its Limitations and Their Causes

The PuC technique of Hammerstein model identification relys on the functions $g_i(t)$ extracted from the response of the matched filter to the exponential swept-sine signal input to the nonlinear system. Let us analyze, in detail, the procedure described in Section 2.1 and reformulate it in the frequency domain.

The procedure makes no reference to limitations in the bandwidth of each of the signals involved, implicitly considering them to have infinite bandwidth (or, in the case of sampled signals, bandwidth up to the Nyquist frequency).

This assumption is not fulfilled in practice. Let the nonlinear system be modeled exactly by an Hammerstein model of order OrdMax. If a sinusoidal signal $s(t) = Cos(\phi(t)) = Cos(2\pi\ f_0 t + \varphi_0)$ is used as the input, the output from the model can be represented by the expression:

$$u(t) = \sum_{k=0}^{Ord_{Max}} H_k(\omega)\ [Cos(2\pi\ f_0 t + \varphi_0)]^k \tag{8}$$

where $H_k(\omega)$ is the transfer function on the kth branch of the model, and, for generality, the zero frequency component ($f_{MIN} = 0$) is considered. Each of the powers of the cosine function can be represented as a sum of its harmonics. The amplitude of these harmonics,

some of which have zero amplitude, are given by the $[A^{-1}]$ matrix, and the inverse of the matrix of the coefficients of the Chebyshev polynomials, given here in the OrdMax = 4 case:

$$
\begin{bmatrix} \mathrm{Cos}[\omega_0 t]^0 = 1 \\ \mathrm{Cos}[\omega_0 t]^1 \\ \mathrm{Cos}[\omega_0 t]^2 \\ \mathrm{Cos}[\omega_0 t]^3 \\ \mathrm{Cos}[\omega_0 t]^4 \end{bmatrix} = \begin{bmatrix} A^{-1} \end{bmatrix} \cdot \begin{bmatrix} \mathrm{Cos}[0\,\omega_0 t] = 1 \\ \mathrm{Cos}\,[1\omega_0 t] \\ \mathrm{Cos}\,[2\omega_0 t] \\ \mathrm{Cos}[3\omega_0 t] \\ \mathrm{Cos}[4\omega_0 t] \end{bmatrix} = \begin{bmatrix} 1 & 0 & 0 & 0 & 0 \\ 0 & 1 & 0 & 0 & 0 \\ \frac{1}{2} & 0 & \frac{1}{2} & 0 & 0 \\ 0 & \frac{3}{4} & 0 & \frac{1}{4} & 0 \\ \frac{3}{8} & 0 & \frac{1}{2} & 0 & \frac{1}{8} \end{bmatrix} \cdot \begin{bmatrix} 1 \\ \mathrm{Cos}[1\omega_0 t] \\ \mathrm{Cos}[2\omega_0 t] \\ \mathrm{Cos}[3\omega_0 t] \\ \mathrm{Cos}[4\omega_0 t] \end{bmatrix} \quad (9)
$$

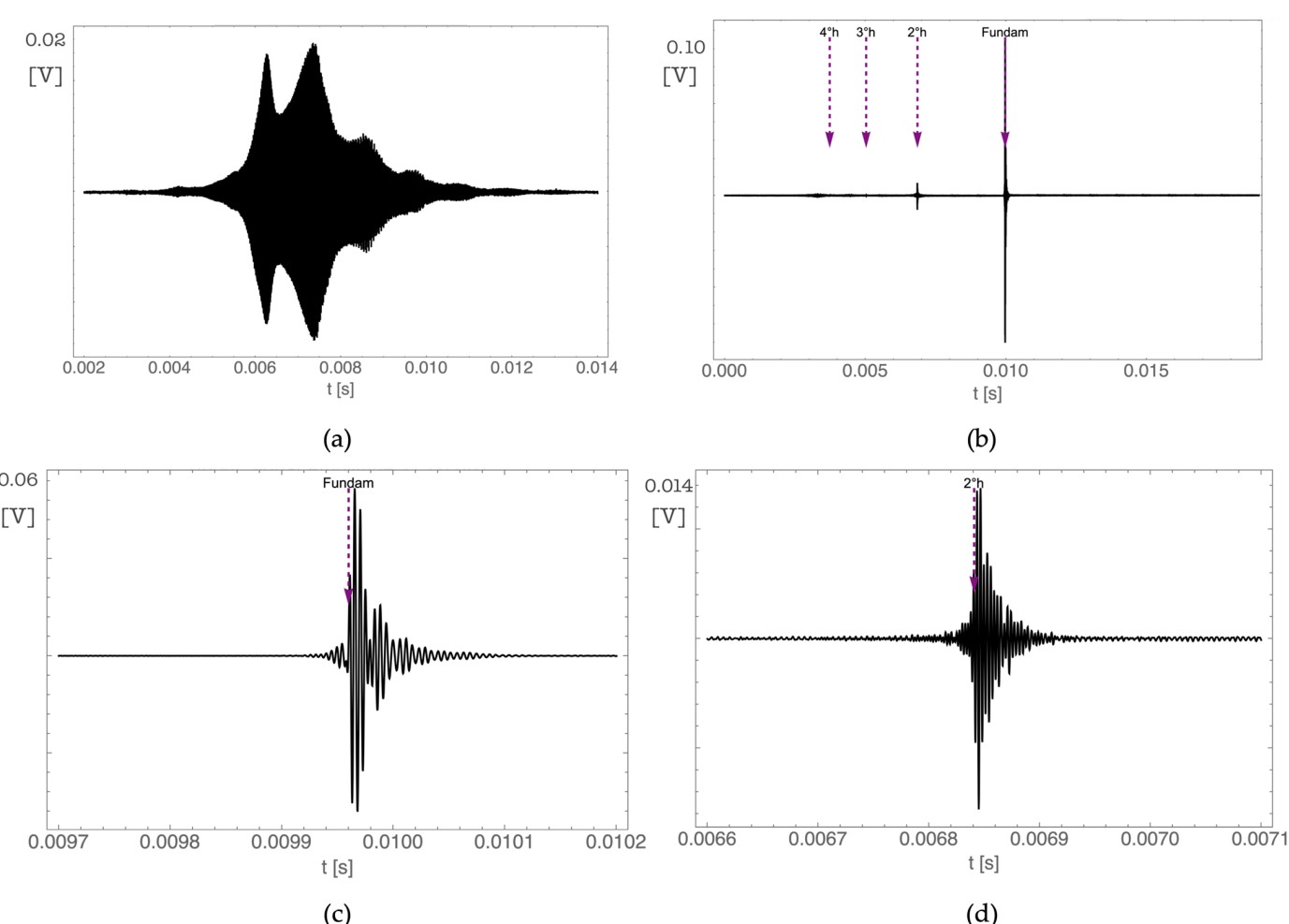

**Figure 5.** The signal at different stages of processing obtained starting from the data measured in the laboratory with the measuring bench described in Figure 4: (**a**) Signal received by the Rx probe; (**b**) output of the matched filter; (**c**,**d**) magnifications of the fundamental frequency and 2° harmonic peaks.

This equivalence makes it possible to give an alternative representation of the output signal u($t$), again of the additive type, in which the sum is extended to harmonics rather than to powers of the input signal. Expanding each power by means of the $[A^{-1}]$ matrix and grouping by harmonics rather than by powers gives:

$$
\mathrm{u}(t) = \sum_{i=0}^{Ord_{Max}} |G_i(i\,\omega_0)|\,Cos(i\,2\pi\,\mathrm{f}_0 t + i\,\varphi_0 + \varphi_i(i\,\omega_0)) \quad (10)
$$

The functions $G_i(\omega)$, whose argument is $\varphi_i(\omega)$, are obtained from the functions $H_k(\omega)$ through the relationship that, again in the OrdMax = 4 case, is:

$$
\begin{bmatrix} G_1(\omega) \\ G_2(\omega) \\ G_3(\omega) \\ G_4(\omega) \end{bmatrix} = \begin{bmatrix} A_C^{-1} \end{bmatrix}^T \cdot \begin{bmatrix} H_1(\omega) \\ H_2(\omega) \\ H_3(\omega) \\ H_4(\omega) \end{bmatrix} = \begin{bmatrix} 1 & 0 & \frac{3}{4} & 0 \\ 0 & \frac{1}{2} & 0 & \frac{1}{2} \\ 0 & 0 & \frac{1}{4} & 0 \\ 0 & 0 & 0 & \frac{1}{8} \end{bmatrix} \cdot \begin{bmatrix} H_1(\omega) \\ H_2(\omega) \\ H_3(\omega) \\ H_4(\omega) \end{bmatrix}
\tag{11}
$$

where superscript T denotes transposed matrix. In this case, the zero frequency component was not considered. The resulting matrix $[A^{-1}]$, deprived of the first row and of the first column, is denoted $[A_C^{-1}]$ as in [19]. The fact that the matrices $[A^{-1}]$ and $[A_C^{-1}]^T$, which appear in Equations (10) and (11) and link the functions $H_k(\omega)$ and $G_i(\omega)$, are upper triangular, has relevant consequences from the examined perspective, as discussed in the following.

All the above considerations apply both in the case where the input signal is a simple cosine function and in the case where the signal is an exponential swept-sine signal, or any other signal characterized by containing, for each instant of time, a single harmonic component.

In the PuC identification technique of the Hammerstein model, the functions $g_i(t)$ are obtained by the convolution between the matched filter and the exponential swept-sine signal at the input, or with its harmonics generated by the nonlinear system. The band in which each $g_i(t)$ is identified is the frequency band common to the i-th harmonic of the exponential swept-sine input signal and the matched filter. The i-th harmonic of the input signal, under the ideal assumption of product T x B tending to infinity, has components between $i*f_{MIN}$ and $i*f_{MAX}$. The frequency band in which the $g_i(t)$ is defined is, thus, related to the choices made for the bands of the input signal and those of the corresponding matched filter. Figures 6–8 consider some possible cases of choosing the frequency bands of the input signal and of the matched filter, and graphically depict how these choices affect the frequency band in which the $g_i(t)$ functions are identified, in the case where they are estimated by the pulse compression technique. First, let us consider the case in which both the swept-sine signal and the matched filter are defined between $f_{MIN}$ and $f_{MAX}$; the situation will be that which, in the case of model order 4, is described in Figure 6. The matched filter, defined between $f_{MIN}$ and $f_{MAX}$, is represented by the dotted curve in the four correlation lags in which it fits the fundamental frequency, and the second, third, and fourth harmonics. It is evident that, in this manner, the function $g_i(t)$ is represented only up to the frequency $f_{MAX}$ whatever harmonic $i$ is considered. It can then be assumed that, if the function $g_i(t)$ being approximated has significant components of the amplitude response beyond $f_{MAX}$, they will not be identified; by adopting this implementation of the PuC technique, there will be a sudden reduction in the amplitude of the identification result obtained as an approximation of the function $G_i(\omega)$.

Figure 7 represents the superposition between the harmonics and the matched filter in the case where the swept-sine signal is defined between $f_{MIN}$ and $f_{MAX}$ and the matched filter is defined in the frequency range from $f_{MIN}$ to $OrdMax * f_{MAX}$. Figure 7 evidences that the frequency bands in which the different $g_i(t)$ functions can be identified are different from each other, with each having the frequency $i * f_{MAX}$ as its upper extreme. It follows that, as shown below, when these $g_i(t)$ functions are adopted to reconstruct the $h_k(t)$ functions that identify the Hammerstein model, the $g_i(t)$ functions combine only partially in the higher frequency ranges, and this is a limitation in the quality of the reconstruction of the $h_k(t)$ functions. This aspect is all the more significant the lower the order k of the $h_k(t)$ function, since, as noted before, the fact that the reconstruction matrices are upper triangular implies that the lower the order k considered for the $H_k(\omega)$, the greater the number of $g_i(t)$ functions that must compose, and in the case that their frequency bands are not the same, it leads to severe limitations in the quality of the result. Finally, Figure 8 shows the situation of the superposition between harmonics and the matched filter in the case in which both the matched filter and the exponential swept-sine input signal are

defined in the frequency range from $f_{MIN}$ to $OrdMax * f_{MAX}$, i.e., their bandwidths are commensurate with the order of the model adopted. All $g_i(t)$ functions are defined in the same frequency band between $f_{MIN}$ and $OrdMax * f_{MAX}$.

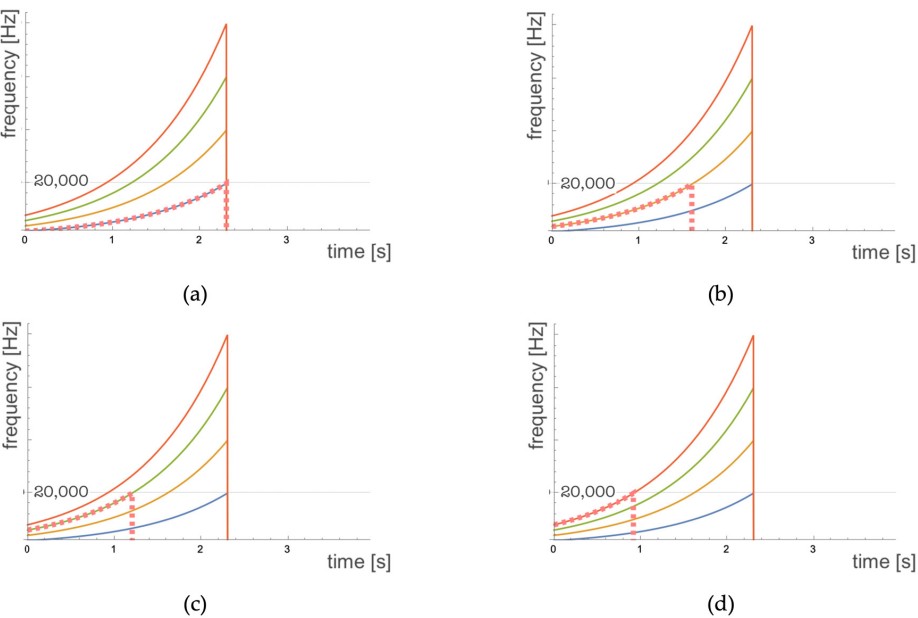

**Figure 6.** Frequency overlap in the case in which both the swept-sine signal and the matched filter are defined between $f_{MIN}$ and $f_{MAX}$. The solid lines represent the swept-sine signal (blue line) its second harmonic (orange line) its third harmonic (green line) and its fourth harmonic (red line). The dotted line represents the frequency band covered by the adapted filter. The four panels represent the cases when the matched filter correlates with (**a**) the fundamental frequency; (**b**) the second harmonic; (**c**) the third harmonic; (**d**) the fourth harmonic.

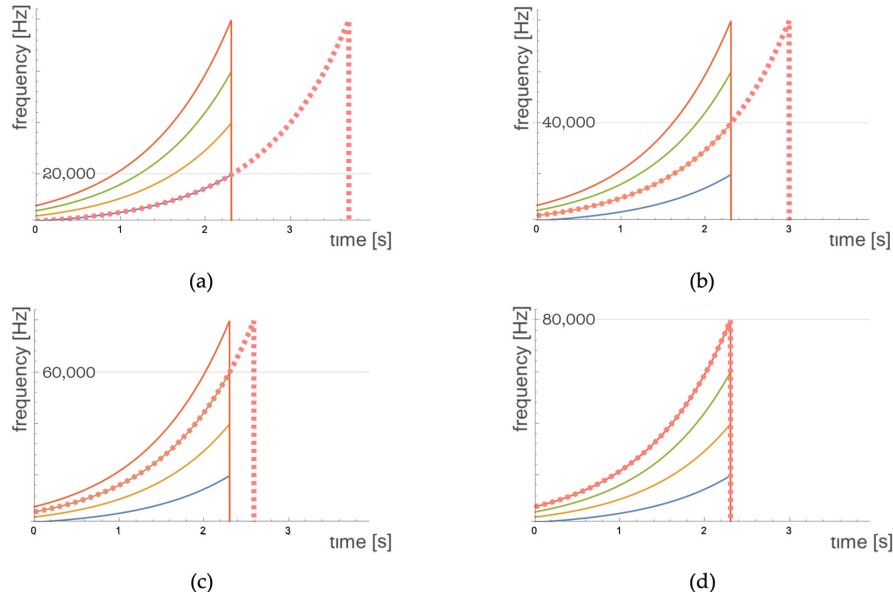

**Figure 7.** Frequency overlap in the case in which the Swept-Sine signal is defined between $f_{MIN}$ and $f_{MAX}$ and the matched filter is defined between $f_{MIN}$ and $OrdMax * f_{MAX}$. The solid lines represent the swept-sine signal (blue line) its second harmonic (orange line) its third harmonic (green line) and its fourth harmonic (red line). The dotted line represents the frequency band covered by the adapted filter. The four panels represent the cases when the matched filter correlates with (**a**) the fundamental frequency; (**b**) the second harmonic; (**c**) the third harmonic; (**d**) the fourth harmonic.

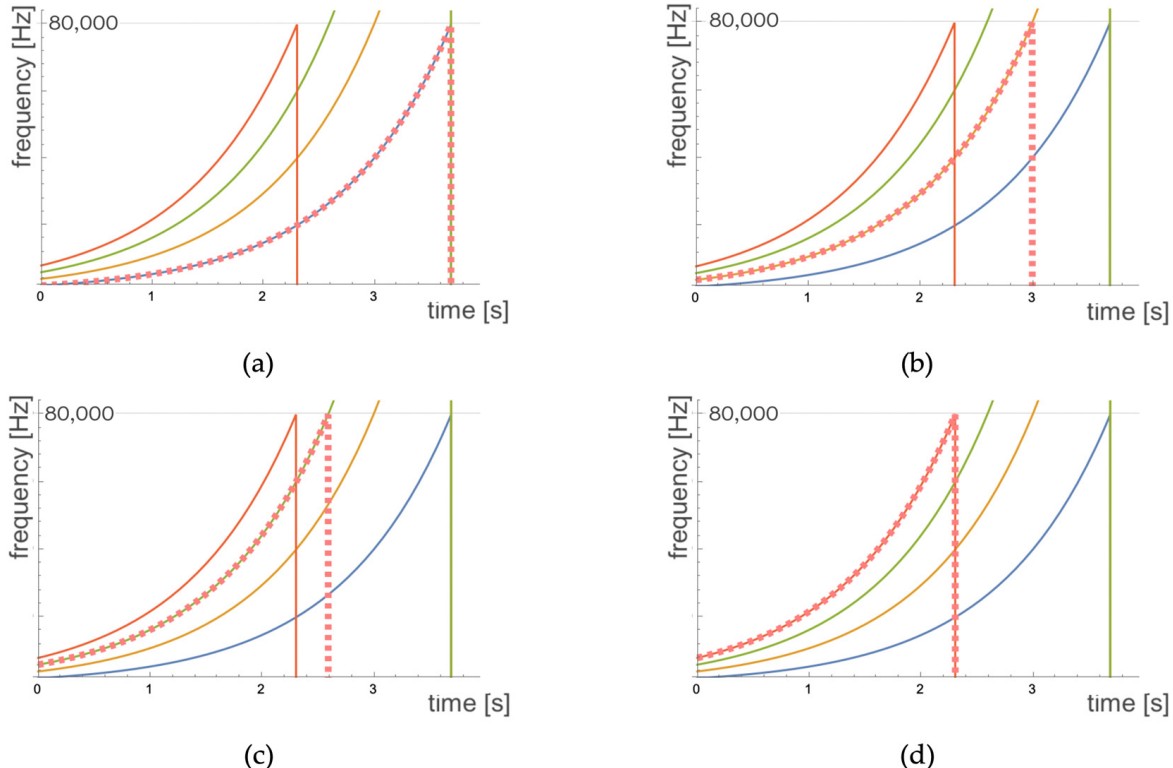

**Figure 8.** Frequency overlap in the case in which both the matched filter and the input swept signal range between $f_{MIN}$ and $OrdMax * f_{MAX}$. The solid lines represent the swept-sine signal (blue line) its second harmonic (orange line) its third harmonic (green line) and its fourth harmonic (red line). The dotted line represents the frequency band covered by the adapted filter. The four panels represent the cases when the matched filter correlates with (**a**) the fundamental frequency; (**b**) the second harmonic; (**c**) the third harmonic; (**d**) the fourth harmonic.

In the next section, an experiment is defined to analyze what impact these limitations of the frequency bands in which the different functions $g_i(t)$ are defined have on the estimation of the functions $h_k(t)$ or $H_k(\omega)$, that is, on the identification of the Hammerstein model.

## 3. Results

The consequences of the aspects that were highlighted in the previous section need to be verified experimentally. To this end, a specific synthetic experiment, described later in this section, was defined to verify the correctness of the observations made in the previous section regarding the bandwidth of the signals involved in the identification step carried out by means of the PuC technique, and to analyze what consequences the choices made on the frequency bands of these signals have on the quality of the Hammerstein model identification result. The choice was made that the simulated nonlinear physical system defined for the experiment was constructed following the exact Hammerstein model. The reason for this choice is that, as a result, the reference transfer functions, $H_k(\omega)$, of each branch, or, equivalently, the corresponding functions, $h_k(t)$, are known to us. The ideal parameters to which the identification procedure should strive are known. The PuC identification technique is applied to this simulated physical system in order to obtain, through the identification procedure, an estimate of the different branch functions of the model. Knowledge of the ideal reference trend of the $H_k(\omega)$ or $h_k(t)$ functions allows us to compare these reference trends with the corresponding trends obtained through the identification technique. Thus, it is possible to verify the impact of different choices of the parameters of the identification procedure on the quality of the result obtained.

The nonlinear physical system was defined by simulating it through a Hammerstein structure. Then, since the ideal functions $h_k(t)$ were known, it was possible to calculate the

output corresponding to the exponential swept-sine signal at the input. Through the PuC technique, the $\widetilde{g}_i(t)$ functions were estimated, their equivalent $\widetilde{G}_i(\omega)$ functions evaluated in the frequency domain and, from these, the $\widetilde{H}_k(\omega)$ functions (and thus the $\widetilde{h}_k(t)$). The PuC estimate of the ideal $H_k(\omega)$ is obtained through Relation (12), which is the inverse of Relation (11):

$$
\begin{bmatrix} H_1(\omega) \\ H_2(\omega) \\ H_3(\omega) \\ H_4(\omega) \end{bmatrix} = \begin{bmatrix} A_C^T \end{bmatrix} \cdot \begin{bmatrix} G_1(\omega) \\ G_2(\omega) \\ G_3(\omega) \\ G_4(\omega) \end{bmatrix} = \begin{bmatrix} 1 & 0 & -3 & 0 \\ 0 & 2 & 0 & -8 \\ 0 & 0 & 4 & 0 \\ 0 & 0 & 0 & 8 \end{bmatrix} \cdot \begin{bmatrix} G_1(\omega) \\ G_2(\omega) \\ G_3(\omega) \\ G_4(\omega) \end{bmatrix} \tag{12}
$$

A comparison of the amplitude responses of the estimated $\widetilde{H}_k(\omega)$ functions obtained from $\widetilde{G}_i(\omega)$ by applying (12) on the results obtained in simulation, comparing the moduli, and the ideal $H_k(\omega)$ or a comparison of the corresponding time functions $h_k(t)$ and $\widetilde{h}_k(t)$ give us an indication of the effects of the choice of parameters adopted in the PuC identification technique.

The system considered in our simulation experiment is a fifth-order Hammerstein system at the input of which an exponential swept-sine signal was input, sampled at 300 kHz, extending in the frequency range from 300 Hz to 10 kHz, with a growth rate L = 0.16583. Consequently, the exponential swept-sine signal duration from $f_{MIN}$ to $f_{MAX}$ is calculated to be T = 0.5815 [s] while the duration up to the frequency OrdMax$*f_{MAX}$ = 50 kHz is T = 0.8484 [s].

On the five branches of the nonlinear system simulated through a Hammerstein model, bandpass filters of different types and order were placed: on branches 1, 3, and 5, Butterworth-aligned bandpass filters of orders 6, 4, and 2, respectively; on branches 2 and 4, Chebyshev-aligned bandpass filters of orders 5 and 3, respectively. The cutoff frequencies of all five bandpass filters are 300 Hz and 20 kHz for the lower and upper cutoff frequencies; the high end cutoff frequency is twice $f_{MAX}$, therefore, that there are significant components of the amplitude responses up to the frequency OrdMax$*f_{MAX}$. All parameters used to define the simulated system in accordance with Hammerstein model are given in Table 1. The simulations were carried out using the software Mathematica™.

**Table 1.** Filter parameters in the five branches of the nonlinear system simulator.

| Model Order | Filter Name | Filter Alignment | Filter Order | Low End Cutoff (Hz) | High End Cutoff (Hz) |
|---|---|---|---|---|---|
| 1 | $H_1(\omega)$ | Butterworth | 6 | 300 | 20.000 |
| 2 | $H_2(\omega)$ | Chebyshev | 5 | 300 | 20.000 |
| 3 | $H_3(\omega)$ | Butterworth | 4 | 300 | 20.000 |
| 4 | $H_4(\omega)$ | Chebyshev | 3 | 300 | 20.000 |
| 5 | $H_5(\omega)$ | Butterworth | 2 | 300 | 20.000 |

Figure 9 shows, on the first line, the five amplitude responses of the filters that were inserted into the branches of the Hammerstein structure that represent the nonlinear system in the simulation. The second line of Figure 9 shows the corresponding five impulse responses. The curves on the two rows are our references in the frequency and time domains, respectively.

Figure 10 compares the results obtained by applying the PuC procedure of model identification, viewed in terms of estimating the frequency responses of the filters on the five branches of the model itself, according to the three different procedures described above for the choice of frequency bands adopted for the exponential swept-sine signal at the input and for the corresponding matched filter. The three procedures can be summarized as follows: All have as their starting point the frequency range from $f_{MIN}$ to $f_{MAX}$, within which the nonlinear system must operate. Specifically, for Procedure 1, the exponential swept-sine input test signal is defined in the range from $f_{MIN}$ to $f_{MAX}$ and the matched filter operates in the same frequency range; for Procedure 2, the range of the exponential

swept-sine input test signal goes from $f_{MIN}$ to $f_{MAX}$, while the matched filter covers the frequency range from $f_{MIN}$ to $OrdMax * f_{MAX}$; for Procedure 3, both the exponential swept-sine test signal at the input and the impulse response of the matched filter cover the frequency range from $f_{MIN}$ to $OrdMax * f_{MAX}$.

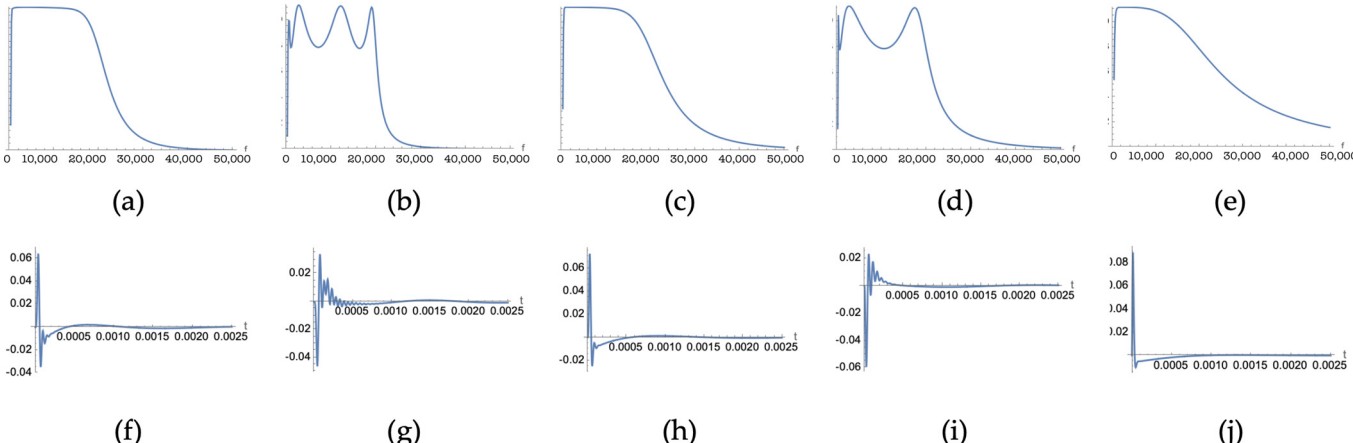

**Figure 9.** Panels from (**a**–**e**) show the amplitude responses of the filters on the five branches of the Hammerstein model considered as reference behaviors in the experiment. Panels from (**f**–**j**) show the corresponding reference impulse responses.

In the case of Procedure 1 (panels "f" to "j"), it is evident that the amplitude response estimation is strictly limited to the maximum frequency $f_{MAX}$ (10 kHz in the experiment) common to the exponential swept-sine signal and the matched filter, and this is in line with what was assumed following the observation of Figure 6.

In the case of Procedure 2 (panels "k" to "o"), the trends exhibit discontinuities at frequency multiple integers of $f_{MAX}$. For example, panel "k" shows a discontinuity at $f_{MAX}$ (10 kHz) and an additional discontinuity at $3f_{MAX}$ (30 kHz). The trends between one discontinuity and the next are not in line with the desired ideal trend of the amplitude response (first row). This result confirms two aspects highlighted earlier. It confirms the considerations made in correspondence to Figure 7, which show that, in this case, the different functions, $G_i(\omega)$, are characterized in frequency bands that are different from each other, and therefore, only up to $f_{MAX}$ are all the components that must add up present, while in higher frequency bands the lower order functions $G_i(\omega)$ are not defined and the combination occurs in the absence of some contributions that would be essential. A second aspect that is evidenced by examining the panels from "k" to "o," is that only some of the components enter into the combination, as evidenced by the structure of matrix $[A_C^T]$ in Relation (12). The $G_i(\omega)$ functions that combine to give rise to the first order function, $H_1(\omega)$, are all the $G_i(\omega)$ components of odd index starting from $G_1(\omega)$, and thus, the frequencies at which the components will be involved are odd integer multiples of $f_{MAX}$ (10 kHz, 30 kHz, ... ); for $H_k(\omega)$ of higher orders, given the upper triangular structure of the $[A_C^T]$ combination matrix, the $G_i(\omega)$ functions that will combine will be those of indices k to OrdMax, and therefore, discontinuities in the combination will occur at higher frequencies. For example, in $H_2(\omega)$, the first discontinuity occurs at $2f_{MAX}$ (20 kHz) and the next discontinuity occurs at $4f_{MAX}$ (40 kHz); in the case of $H_3(\omega)$, the first discontinuity is found at $3f_{MAX}$ (30 kHz) and the next discontinutiy is found at $5f_{MAX}$ (50 kHz).

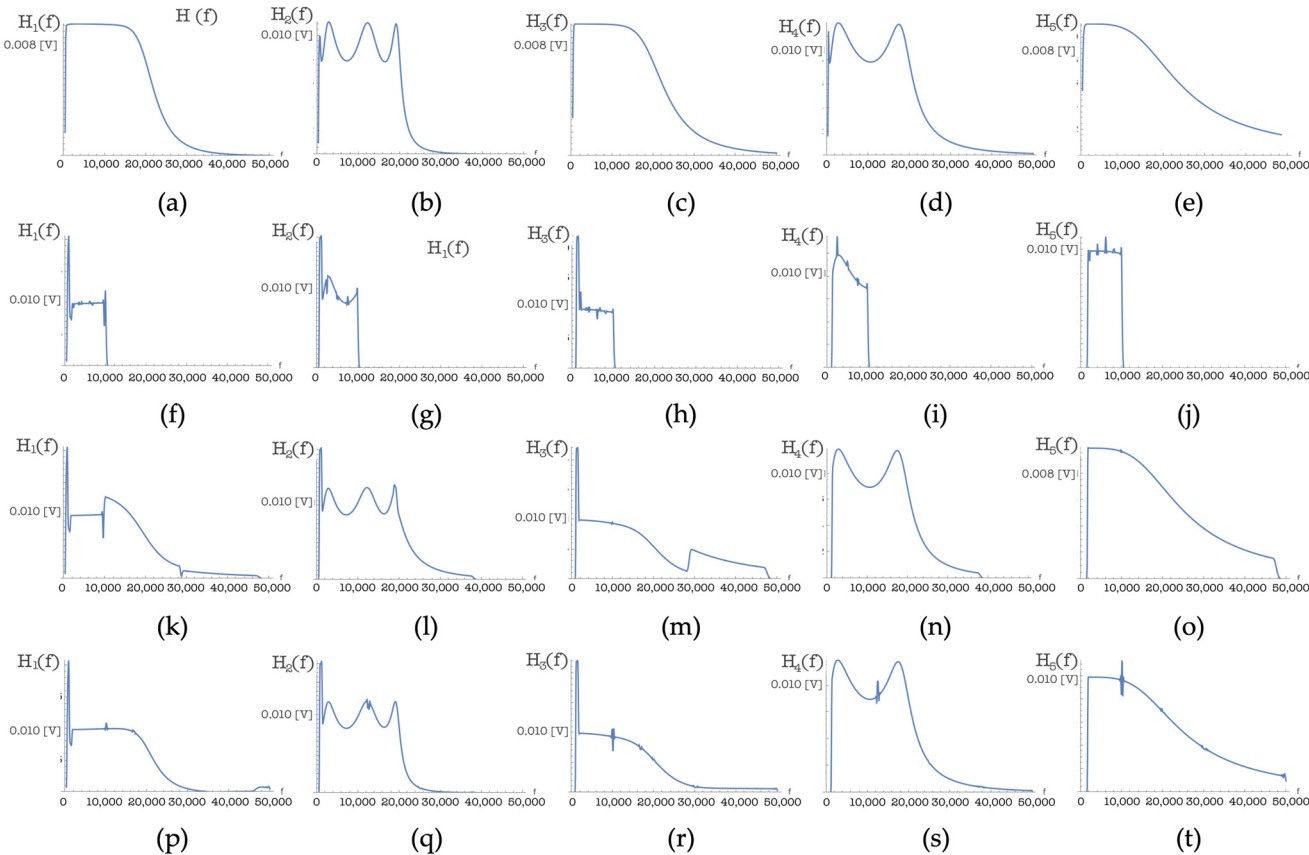

**Figure 10.** Panels from (**a**–**e**) show the reference amplitude responses of the filters on the five branches of the Hammerstein model in the experiment. Panels from (**f**–**j**) show the amplitude responses estimated with the Procedure 1 (swept-sine range from $f_{MIN}$ to $f_{MAX}$; matched filter range from $f_{MIN}$ to $f_{MAX}$). Panels from (**k**–**o**) show the result applying Procedure 2 (swept-sine range from $f_{MIN}$ to $f_{MAX}$; matched filter range from $f_{MIN}$ to $OrdMax * f_{MAX}$). Panels from (**p**–**t**) show the amplitude responses estimated with the Procedure 3 (swept-sine range from $f_{MIN}$ to $OrdMax * f_{MAX}$; matched filter range from $f_{MIN}$ to $OrdMax * f_{MAX}$).

In the case of identification Procedure 3 (panels "p" to "t"), the trends faithfully reflect throughout the frequency band of interest the ideal trends. It is evident that the result comes from the combination of several components and, in fact, at frequency multiples of $f_{MAX}$, small irregularities are found; however, the overall trend is consistent with the ideal reference, and this denotes that there is a correct combination of all the necessary components. The only aspect that merits some further investigation concerns the peak found at the minimum frequency $f_{MIN}$.

The corresponding result obtained in the time domain is very interesting. For ease of reading, in each of Figures 11a, 12a and 13a are reported the reference trend of each of the three functions $h_1(t)$, $h_2(t)$, and $h_3(t)$ considered, respectively, shown as compared with those obtained by the identification Procedures 1, 2, and 3, plotted in Figures 11b–d, 12b–d and 13b–d, respectively.

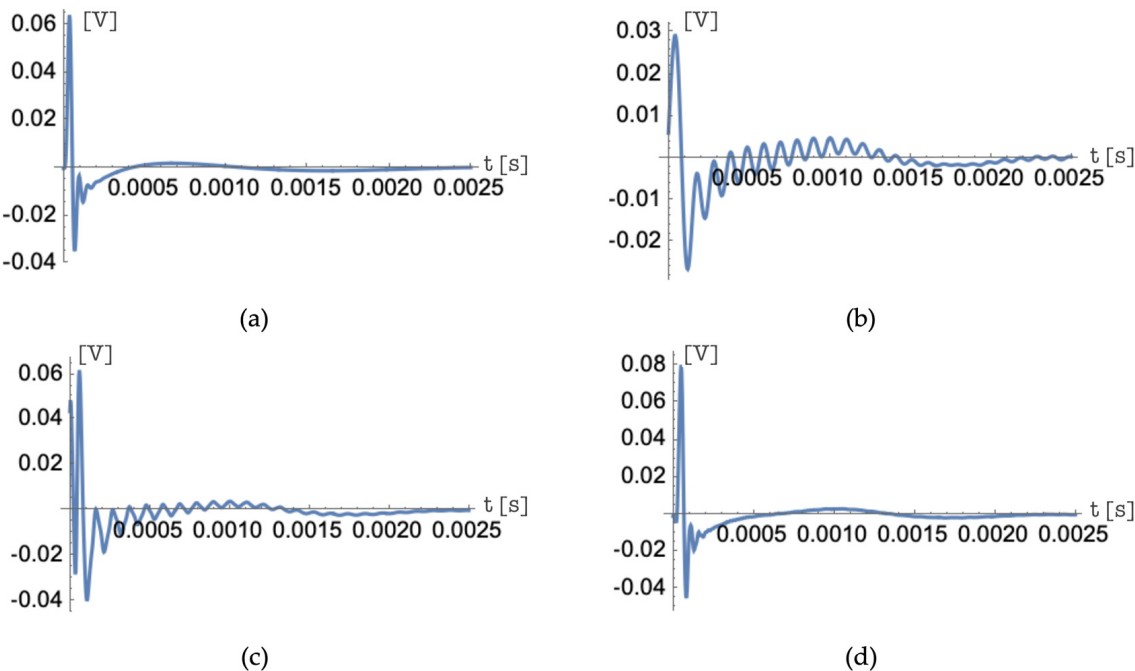

**Figure 11.** Function $h_1(t)$: Comparison of (**a**) the reference trend with those obtained by (**b**) identification Procedure 1 (swept-sine range from $f_{MIN}$ to $f_{MAX}$ and matched filter range from $f_{MIN}$ to $f_{MAX}$); (**c**) identification Procedure 2 (swept-sine range from $f_{MIN}$ to $f_{MAX}$ and matched filter range from $f_{MIN}$ to $OrdMax * f_{MAX}$); (**d**) identification Procedure 3 (swept-sine range from $f_{MIN}$ to $OrdMax * f_{MAX}$ and matched filter range from $f_{MIN}$ to $OrdMax * f_{MAX}$).

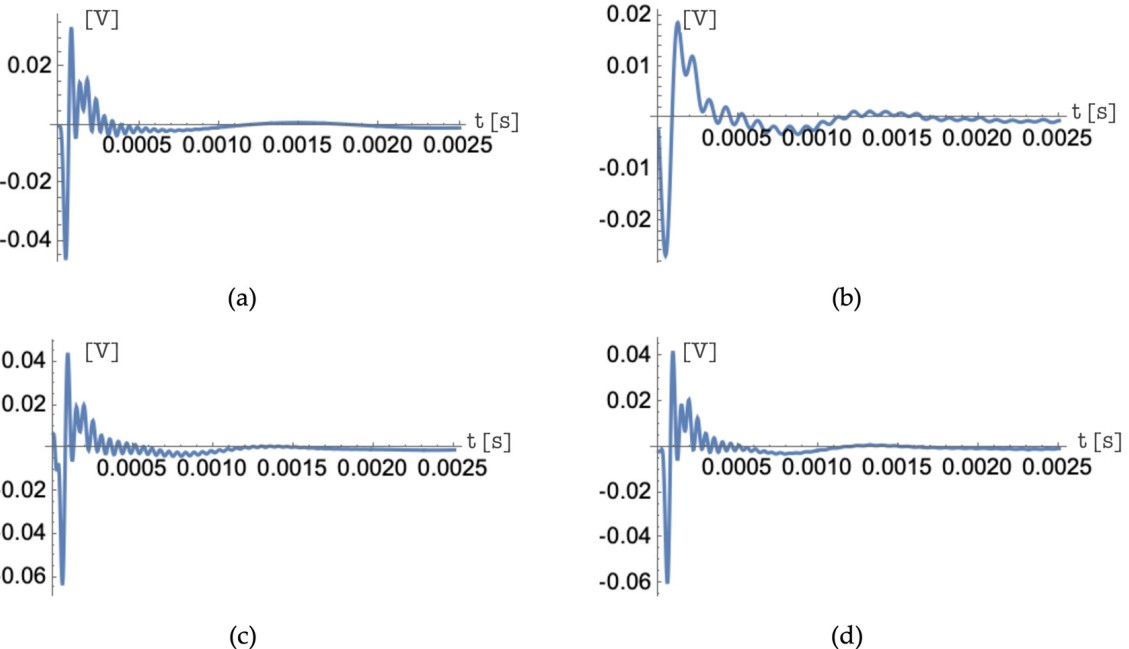

**Figure 12.** Function $h_2(t)$: Comparison of (**a**) the reference trend with those obtained by (**b**) identification Procedure 1 (swept-sine range from $f_{MIN}$ to $f_{MAX}$ and matched filter range from $f_{MIN}$ to $f_{MAX}$); (**c**) identification Procedure 2 (swept-sine range from $f_{MIN}$ to $f_{MAX}$ and matched filter range from $f_{MIN}$ to $OrdMax * f_{MAX}$); (**d**) identification Procedure 3 (swept-sine range from $f_{MIN}$ to $OrdMax * f_{MAX}$ and matched filter range from $f_{MIN}$ to $OrdMax * f_{MAX}$).

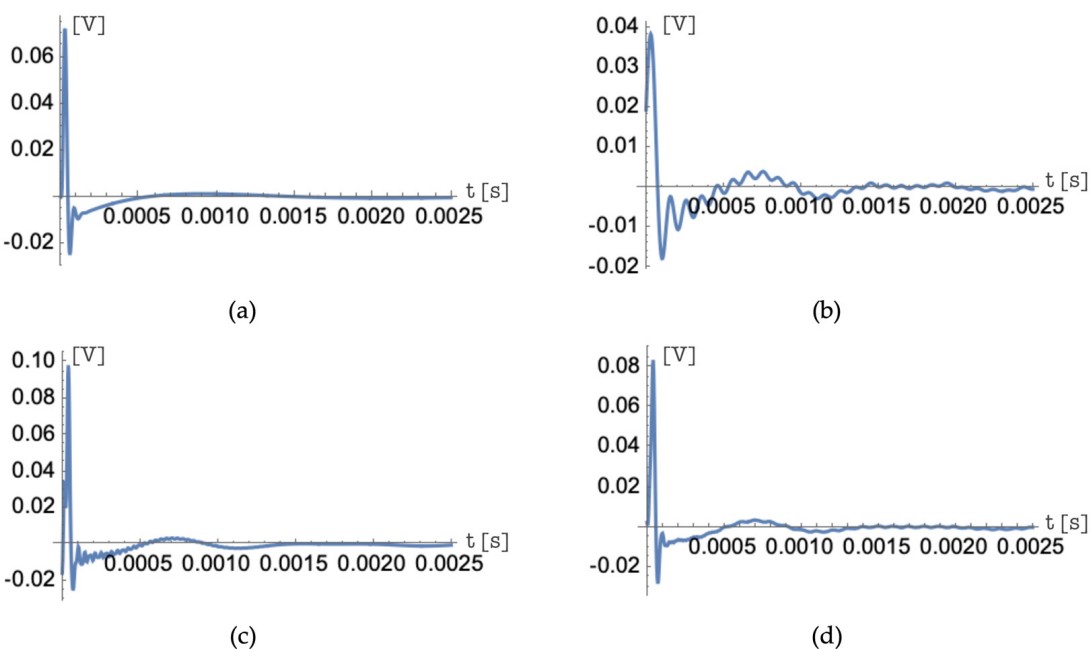

**Figure 13.** Function $h_3(t)$: Comparison of (**a**) the reference trend with those obtained by (**b**) identification Procedure 1 (swept-sine range from $f_{MIN}$ to $f_{MAX}$ and matched filter range from $f_{MIN}$ to $f_{MAX}$); (**c**) identification Procedure 2 (swept-sine range from $f_{MIN}$ to $f_{MAX}$ and matched filter range from $f_{MIN}$ to $OrdMax * f_{MAX}$); (**d**) identification Procedure 3 (swept-sine range from $f_{MIN}$ to $OrdMax * f_{MAX}$ and matched filter range from $f_{MIN}$ to $OrdMax * f_{MAX}$).

The considerations can be made cumulatively with reference to the three figures considered. Identification Procedure 1, which involves a significant limitation in the bandwidth of the identified function and an abrupt jump in the amplitude response at $f_{MAX}$, leads, in all cases considered, to an increase in the duration of the initial peak and to a strongly oscillating trend in the response over time.

In the case of identification Procedure 2, it is not so much the band limitation that makes its effects on the time course, but the irregularities, which are observed in the amplitude response, are reflected in irregularities in the time course and, again, in oscillations in the response.

Only identification Procedure 3, in all the cases considered, provides an adequate ability to regain the time course of the functions considered, and thus, a correct identification of the Hammerstein model in both the time and frequency domains. It can be added that the trends of functions $h_4(t)$ and $h_5(t)$ lead to essentially equivalent considerations to those of the three functions $h_1(t)$, $h_2(t)$, and $h_3(t)$ shown in Figures 11–13, respectively.

## 4. Discussion

The present work addressed the problem of the quality of the estimation of kernels characterizing the different branches of a Hammerstein model of a nonlinear system, in the case in which such a model is identified through the PuC technique. First, the work was concerned with verifying the real existence of the problem and, in particular, the presence of spurious oscillations at transitions in the time response of the system. To do this, and to verify that this problem is present in real physical devices, an attempt was made to verify the existence of the problem through a laboratory experiment. The experiment involved an ultrasonic system with probes designed to operate in air. An analysis of the results obtained by modeling the real physical system through a Hammerstein model showed that the impulsive responses of the various filters entering into the model's characterization do indeed exhibit precursors, in the form of oscillations that anticipate the theoretically calculated instant of attack. It was hypothesized that the presence of such oscillations was associated with the Gibbs phenomenon, which motivated us to analyze possible limitations

of the frequency band covered by the characterization of the different impulse responses of the filters that constitute the kernels of the Hammerstein model. This analysis of the frequency bands was carried out by rewriting the PuC procedure in the frequency domain and observing how the final result, i.e., the $H_k(\omega)$ functions, were obtained by linearly combining the $G_i(\omega)$ functions, which, in turn, were obtained through the convolution between the response of the nonlinear system to an exponential swept-sine signal and the filter matched to that signal.

Because of the way the matched filter was defined in the present case (optimized for additive white Gaussian noise), the convolution with the matched filter was equivalent to the correlation function of the response of the nonlinear system with the signal at its input.

If the signal at the input and the matched filter both had infinite bands, the correlation would extend over the entire frequency range; since the bands are limited, the frequency band in which the correlating signals overlap, i.e., the band in which the procedure allows the definition of the $G_i(\omega)$ functions, depends on the frequency bands in which the input signal and the matched filter are defined. In addition, it can be observed that the combination of $G_i(\omega)$ functions occurs through matrices whose upper triangular structure means that the $H_k(\omega)$ functions of low $K$ index, which are in most cases energetically more significant, are obtained by combining more $G_i(\omega)$ functions than for the $H_k(\omega)$ functions of high $K$ index. This implies that if $G_i(\omega)$ functions are defined in frequency bands that are inconsistent with each other, the $H_k(\omega)$ functions that will be most affected will be those that are energetically more significant. Figures 6–8 graphically represent the effect of frequency band limitations in the estimation of $G_i(\omega)$ functions. An analysis of the schematizations shown in these figures shows that in order for the $G_i(\omega)$ functions to ensure that the frequency band covered by the correlation, for all $K$ indices, is the necessary one, both the exponential swept-sine signal as input and the matched filter must be defined in the frequency band from $f_{MIN}$ to $\text{OrdMax} * f_{MAX}$.

To verify that the analysis performed was correct, a specific experiment was designed. The experiment had to be able to compare the result obtained through PuC identification with the ideal result. To have such an ideal reference available, we chose to adopt a simulated experiment in which the nonlinear system was realized according to the Hammerstein model scheme. In this way, what kernel trends are expected to be estimated by the PuC identification system were known. The Results section describes the nonlinear system simulator and the results of the identification by means of the PuC procedure. The simulator was realized using a fifth-order Hammerstein system, placing on the five branches in parallel, linear filters whose parameters are given in Table 1, and whose amplitude and impulse responses are shown in Figure 9. Figure 10 shows, in terms of the amplitude responses of the five filters, the comparison between the ideal trends and the trends obtained by adopting the PuC procedure with three different choices of the frequency bands of the input signal and the matched filter. The reported result highlights the correctness of the assumptions made about the necessary bands. Only in the case in which both the exponential swept-sine signal at the input and the corresponding matched filter have frequency band ranging from $f_{MIN}$ to $\text{OrdMax} * f_{MAX}$, are all five amplitude response trends obtained by PuC identification reasonable estimates of the ideal trends across the whole useful band.

Figures 11–13 show similar results in the time domain, comparing, in each figure, the ideal impulse responses and those obtained with the three choices of the frequency bands of the input signal and of the corresponding impulse response of the matched filter. Figures 11–13 show the comparison of the results over time for the impulse responses $h_1(t)$, $h_2(t)$, and $h_3(t)$, respectively. The time-domain analysis also confirms all the assumptions made. The best approximation always occurs when the chosen frequency bands range from $f_{MIN}$ to $\text{OrdMax} * f_{MAX}$ and, in the case of different choices for frequency bands, the trends of the lowest index $h_k(t)$ functions are most strongly affected by this frequency limitation.

## 5. Conclusions

The present work focused on the quality of Hammerstein model identification, especially in the case in which high quality identification in the time domain is of interest. In this paper, we focused on an aspect observed in experiments, but on which the technical literature had not dwelt, i.e., the frequency band of the impulsive signals that are estimated through the PuC procedure as kernels that characterize the Hammerstein model. The analysis performed showed that the frequency band of the estimated signals was directly related to the frequency band of the exponential swept-sine test signal placed as input in the identification phase and to the band of the impulse response of the matched filter corresponding to that input signal. In the real case, both the test signal placed as input and the filter matched to that input signal are characterized by limited frequency bandwidths. This limitation is reflected in limitations that may not be homogeneous in the frequency bands of the impulsive signals being estimated.

The results obtained confirmed all the assumptions made in the paper. It is intended to continue research in this area to clarify the reason for the peak found at the minimum frequency, $f_{MIN}$, of the estimated kernels. In addition, we paln to extend this research activity along two lines: The first extension is to apply this criterion for choosing signal bands to some of the applications already addressed with Hammerstein models identified through the PuC technique. The second line of evolution of the research activity will be to test the possibility of defining procedures to improve the kernel estimation where, for practical reasons, it is not possible to extend the bandwidth of the input signal or of the matched filter to the optimal limit, and therefore, to prevent the effects of bandwidth limitations from being uncontrolled, and to test whether it is possible to limit negative effects on model identification.

**Funding:** The author acknowledges the support of the "Fondazione Cassa di Risparmio di Terni CARIT" for the partial financing of the research through the project "HydroTOUR—Hydrogen Terni Orizzonte Università e Ricerca".

**Institutional Review Board Statement:** Not applicable.

**Informed Consent Statement:** Not applicable.

**Data Availability Statement:** The data underlying the results presented in this paper are not publicly available at this time but may be obtained from the author upon reasonable request.

**Conflicts of Interest:** The authors declare no conflict of interest.

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
