# Peer review of "Parameter Optimization for an Accurate Swept-Sine Identification Procedure of Nonlinear Systems"

_applsci, doi:10.3390/app13021223_

Round 1

Reviewer 1 Report

1. The authors discussed the impact of the bandwidth in Hammerstein model identificaion. The analysis is straightforward and the experiments results supported the authors' opinion apparently.

2. The figures in this article are recommended to be reviesed more clearly.  The kernels "h(t)" in Figure 1 seem to have some mistakes .  The four curves in Figures 4, 5 and 6 will be  easier to be understood if they are shown seperately.  The results in Figures 7 and 8 also seem much indistinct to be observed. 

3. The experiment should be introduced more adequately.  The results in Figures 3 and 7 are shown but not explained in detail.  It will be helpful to understood the experiments results if the author added more information about the measurement system.  

Author Response

Please find enclosed in the attached pdf file:
-a cover letter to explain, point by point, the details of the revisions to the manuscript and our responses to the referees' comments
-the result of applying the function "Track Changes", as requested
-a copy of the new version of the manuscript

Sincerely
Pietro Burrascano

Reviewer 2 Report

Authors of the work entitled “Parameters Optimization for an accurate Swept Sine Identification Procedure of Nonlinear Systems” present a proposal for analysis of the role played by the working bandwidth of both the signal that is used as input in a system, as well as that which belongs to the compatible filter used. Through the analysis of nonlinear systems carried out using the Hammerstein model, and Exponential Swept Sine type signals, a criterion is proposed to define the adequate bandwidth to maximize the accuracy in the operation of the system and the parameters that compose it. The work has academic value and the results are interesting. However, the presentation and the structure of the manuscript hinders its reading, making it difficult to actually understand the contribution of the manuscript. The manuscript could be publishable in Applied Sciences only if the following major comments could be addressed.

Abstract section

– It is suggested to start with a brief explanation about the use and application of nonlinear systems. Likewise, it is suggested that prior to the mention of the Hammerstein model, that there are other models and techniques for identifying parameters for systems, since the way in which the abstract begins is very direct, and does not give the reader a context about the origin of the research and the setting in which it develops.

– Once the previous comment has been addressed, it is appropriate to place lines 8 and 9 of the article.

– It is recommended to add between lines 11 and 12 a brief clarification mentioning that since for different types of systems and their applications, the design elements or parameters of which they are composed are important. The above because the next part of the section deals with bandwidth and the fact that the article is focused on said parameter. It is also recommended to emphasize the fact that bandwidth is a fundamental parameter.

It is recommended to briefly specify what assumptions are made about the amplitudes in frequency and why it is important to verify these assumptions in a way that is useful for the characterization of the system.

Introduction section

– In lines 27, 36 and 37 it is recommended to add references where the white-box, grey-box and black-box concepts are described, since only references to examples of systems of these types are placed, however, it is considered It is important to add a reference that describes the concept. Likewise, since this article focuses on the study of nonlinear systems and begins with the description of the types of systems, it is recommended to add more references where examples, applications, case studies, or other articles where use is made are shown. of these system models.

– In line 37 the concept of black-box and its relevant characteristics is placed, and later the Volterra model is discussed, which allows characterizing this type of system. It is intuitive that the type of system for which the analysis of this work is developed is a black-box system, however, this is not clear. It is strongly recommended to modify the wording so that before beginning to describe characterization techniques or models, the type of system with which one works is clarified.

– Once the previous comment has been addressed, it is suggested, for line 44, in the Hammerstein and Wiener concepts, add a reference for each of the models.

– In the paragraph made up of lines 50 and 54, authors are suggested to place other models that have been found or studied, including those that are mentioned in the abstract, with their respective references.

– On line 62 a reference to Gibbs artifacts is required.

In general, it is necessary to reinforce the state of the art in the introduction, since although, if references to various examples and concepts related to the topic to be developed appear, it is suggested to complement them with references to other similar or related works.

Materials and Methods section

Hammerstein Model and Pulse-Compression identification procedure subsection

– Before assuming the characteristics of the model that is being used, it is suggested to make a brief explanation about what the Hammerstein model is, what is its operation and what is the result that is obtained after applying said model.

– In the paragraph made up of lines 95 to 98, it is recommended to use third person writing at the beginning of the paragraph.

– In the paragraph made up of lines 99 to 103, it is recommended to clarify the difference between the Pulse Compression technique and the Hammerstein model, since the terms model and technique are used interchangeably, which causes confusion within the article when distinguishing the use of each technique. and model.

– In line 114, it is suggested that the authors mention that since the input or excitation signal used is of the Swept Sine type, then the transformation observed in equation 4 can be performed

– A reference is required in line 120.

– After line 147, it is recommended that the authors explain what results are obtained from the model presented in figure 1, that is, that it implies obtaining the functions h_k(t), since it would be appropriate to go to the following sections taking into account clear the result of the Hammerstein model.

Discussion on the Pulse-Compression Identification Procedure subsection

In line 149, a discussion about the PuC technique begins and there is talk of an experiment to verify and analyze the problem presented in the introduction, however, it has not been explained until now what the experiment is about, and immediately a discussion is made about the analysis made. It is recommended that this subsection be modified to explain in principle what is the configuration of the experiment, what is the process, the elements used and the results that are intended to be obtained from it. Said section would also be used to refer to Figure 2 that is presented in this section and is not referenced at any time.

Limitations in the identification procedure subsection

– The Pulse-Compression Identification technique should also be included in the state of the art, with their respective use examples and references.

– It is suggested that the explanation of Figure 2, between lines 174 to 184, be placed in the section where the experiment is described.

– Previously, it was mentioned that the verification to analyze the bandwidth is done through simulation, however, until now the characteristics of said simulation or the results that were intended to be obtained have not been described. If such a simulation was performed, it is recommended to include those results and descriptions in this subsection. It would also be appropriate to mention in which software or tool said simulation was carried out.

– Figure 3 does not allow us to identify whether these results were obtained from the simulation or from the experiment carried out. It is recommended to make a clear difference and explain how each of the results is obtained.

– En la Figura 3, se sugiere agregar las unidades en el eje y, con el fin de especificar las magnitudes que se están comparando.

– Line 167 requires a reference for the concept of estimator integral type estimator.

– In line 177, you must define what the variable F means and its initialization value of 10 cm.

Why is the problem generated? subsection

– It is suggested to modify the title of the subsection for a title that alludes to the transformation of the analysis in the time domain, to the frequency domain.

– It is suggested to specify if the results shown in Figures 4 and 5 were obtained by simulation, or by the physical experiment performed.

– It is important to summarize in a table, or in a figure, the parameters used in the setup of the experiment, since it is mentioned that the article focuses on the optimization of the bandwidth parameter by identifying patterns within a system not linear. It can be deduced that the same values ​​were not used in all the tests carried out. It is recommended to summarize the values ​​of the experiment setup parameters in each test performed.

Results section

The results section would improve its structure and the reading follow-up if all the results obtained are placed here, both those found in the Materials and Methods section and the responses of the filters shown in figures 7, 8, 9 10 and 11, in such a way that section 2 and the subsections that compose it contain the description of the elements used to carry out the work, as well as the summary of the characteristics of the experiments and the types of filters with which the tests were carried out. tests.

– It is recommended to the authors in each presentation of results and graphs, to clarify if these were obtained through simulation or through experimentation and tests.

– It should be clearer with the results obtained regarding the study of the bandwidth, since various tests carried out through the experiment and simulation are shown, however, it is not clear what is the criterion that was obtained to optimize the bandwidth. band and therefore improve the use of the Hammerstein model. Authors are recommended to represent a comparison between a known bandwidth value and an improved one, or those necessary according to the experiment and test carried out.

Discussion and conclusions section

– It is strongly recommended to separate the Discussion section from the Conclusions section, since the work carried out is interesting, one of the most important parameters in the design of systems and signal analysis is studied and therefore several results were obtained.

– In the discussion section it is recommended to place detailed descriptions of the results presented both in graphs and tables and within the text. The discussion of the results presented is very brief and rarely touches on the focus of the work.

In the conclusions section, it is recommended to broaden the description of whether the criterion obtained marks a clear difference between what is used in the literature and the results obtained experimentally and in simulation, in order to compare the results of both approaches (simulation and experimentation). Likewise, it is suggested to locate future work and the state of the research at the time of writing the article.

Best regards!

Author Response

(The authors gave the same response as above.)

Round 2

Reviewer 2 Report

The work entitled "Review Parameters optimization for an accurate Swept Sine Identification Procedure of Nonlinear Systems" was considerably improved since the changes marked by the previous review were made, so in this version the authors delved into the writing of the content and in explanation and presentation of the results. Likewise, it is evident the meticulous dedication that was given to the restructuring of the paper and the attention paid to the writing and format. The manuscript could be accepted after the following minor changes:

Abstract section

– The authors performed the indicated adjustments in prior version. The abstract section was considerably improved, so both the context, purpose and results to obtain are clear.Keywords section

– It is suggested to the authors add some more keywords in order to improve the location of the principal concepts in which the paper is based.Introduction section

– The authors added the requested references. It should be noted that after an exhaustive review, it was verified that these references are scientific articles of reliable origin.

– In lines 50 and 51 it is clarified that the work is based on a black box type system, which allows the reader to remain with that idea during the rest of the reading of the article. However, the authors write said line in the first person plural. Authors are recommended to write in the third person in order to give formality to the writing.

– The authors added the requested references. It should be noted that after an exhaustive review it was verified that these references are scientific articles of reliable origin, however some of them are very old, so it is recommended that the authors update them with more recent works.

– The state of the art has been considerably improved, thus reinforcing the context in which this work is carried out and clarifying the purpose and importance of the study carried out.

Materials and methods section

– In Figure 2, the sentence We note the the time distance between the ... the article the is repeated.

– Authors are requested to add the appropriate reference and permission to use Figure 3

– On line 241, authors are asked to add a reference for the TiePie handyscope HS5 instrument used. This reference must be the original online technical documentation or data sheet of the instrument.

– On line 242, authors are asked to add a reference to Falco System power amplifier. This reference must be the original online technical documentation or data sheet of the instrument.

– Figure 5 requires, for each of the subfigures that axes be added. For the y-axis, it is required to place the units and the dependent variable that is obtained or measured, in addition to the corresponding scale.

– Figure 6,7 and 8 requires, for each of the subfigures, that axes be added. For the y-axis, it is required to place the units and the dependent variable that is obtained or measured, in addition to the corresponding scale.Results section

-- Figures 10, 11 and 12 for each subfigure, require axes to be added. For the y-axis, it is required to place the units and the dependent variable that is obtained or measured, in addition to the corresponding scale.

Regards.

Author Response

Please find enclosed in the attached pdf file:
-a cover letter to explain, point by point, the details of the revisions to the manuscript and our responses to the referees' comments
-a copy of the new version of the manuscript

Sincerely
Pietro Burrascano
